# Proteomics-Based Detection of Immune Dysfunction in an Elite Adventure Athlete Trekking Across the Antarctica

**DOI:** 10.3390/proteomes8010004

**Published:** 2020-03-03

**Authors:** David C. Nieman, Arnoud J. Groen, Artyom Pugachev, Andrew J. Simonson, Kristine Polley, Karma James, Bassem F. El-Khodor, Saradhadevi Varadharaj, Claudia Hernández-Armenta

**Affiliations:** 1North Carolina Research Campus, Appalachian State University, Kannapolis, NC 28081, USA; simonsonaj@appstate.edu; 2ProteiQ Biosciences GmbH, 10967 Berlin, Germany; arnoud@proteiq.com (A.J.G.); artyom@proteiq.com (A.P.); chernand@ebi.ac.uk (C.H.-A.); 3Standard Process Nutrition Innovation, Kannapolis, NC 28081, USA; kpolley@standardprocess.com (K.P.); kjames@standardprocess.com (K.J.); belkhodor@standardprocess.com (B.F.E.-K.); svaradharaj@standardprocess.com (S.V.)

**Keywords:** blood proteins, exercise, immune system, complement, neutrophils, apolipoproteins, nutrition

## Abstract

Proteomics monitoring of an elite adventure athlete (age 33 years) was conducted over a 28-week period that culminated in the successful, solo, unassisted, and unsupported two month trek across the Antarctica (1500 km). Training distress was monitored weekly using a 19-item, validated training distress scale (TDS). Weekly dried blood spot (DBS) specimens were collected via fingerprick blood drops onto standard blood spot cards. DBS proteins were measured with nano-electrospray ionization liquid chromatography tandem mass spectrometry (nanoLC-MS/MS) in data-independent acquisition (DIA) mode, and 712 proteins were identified and quantified. The 28-week period was divided into time segments based on TDS scores, and a contrast analysis between weeks five and eight (low TDS) and between weeks 20 and 23 (high TDS, last month of Antarctica trek) showed that 31 proteins (*n* = 20 immune related) were upregulated and 35 (*n* = 17 immune related) were downregulated. Protein–protein interaction (PPI) networks supported a dichotomous immune response. Gene ontology (GO) biological process terms for the upregulated immune proteins showed an increase in regulation of the immune system process, especially inflammation, complement activation, and leukocyte mediated immunity. At the same time, GO terms for the downregulated immune-related proteins indicated a decrease in several aspects of the overall immune system process including neutrophil degranulation and the antimicrobial humoral response. These proteomics data support a dysfunctional immune response in an elite adventure athlete during a sustained period of mental and physical distress while trekking solo across the Antarctica.

## 1. Introduction

Successful training leading to enhanced performance involves cycles of overload and adequate recovery [1,2,3,4]. A primary goal during training is to avoid the combination of excessive overload and inadequate recovery leading to non-functional overreaching (NFOR) and the overtraining syndrome (OTS) that can result in long term performance decrements and psychological disturbances [1].

Practical and sensitive diagnostic tools are needed to identify athletes with NFOR and OTS, but valid blood biomarkers that can be combined with performance and psychological measurements are lacking. There is a growing interest in the use of protein-based biomarkers for NFOR and OTS because protein–protein interactions are specific, information rich, and biochemically diverse [4,5]. Technological and bioinformatics advances now allow proteomics analysis to be conducted from dried blood spot (DBS) samples with the identification of specific protein patterns that can be linked to underlying biological processes. The use of DBS samples offers many advantages, especially in athletic and military field settings, including ease and safety of transport, storage, and handling [4,6].

Our research group recently utilized a proteomics approach with DBS samples to identify a cluster of 13 proteins that were expressed during two days of recovery from a three day period of intensive exercise [4]. Protein–protein interactions analysis indicated underlying biological processes related to the acute phase response, complement activation, and innate immune system activation. In a subsequent study, this cluster of 13 proteins successfully identified NFOR in an athlete during the Race Across America (RAAM) [7]. The athlete completed the 4941 km race in 10.1 days with only 20 h of sleep, and experienced high psychological training distress and decreased post-race work capacity. Targeted proteomics procedures were conducted on DBS samples that were collected before and after, and twice daily during RAAM, and revealed large fold increases for specific immune-related proteins including complement component C7 (359%), complement C4b (231%), serum amyloid A4 protein (210%), inter-alpha-trypsin inhibitor heavy chain H4 (191%), and alpha-1-antitrypsin (188%). These data are consistent with results from multiple human and animal studies showing that immune-related proteins represent a large proportion of those generated during intensive and prolonged acute and chronic exercise training [8,9,10,11,12,13]. These proteins, when combined with psychological, performance, and nutrition intake data, can serve as useful NFOR and OTS biomarkers of immune dysfunction, training distress, exercise-induced muscle damage and exhaustion, and impaired performance capacity [14,15,16,17,18,19,20,21]. We sought to extend these findings by using proteomics monitoring of an elite adventure athlete over an extended period of time (28 weeks) that culminated in the successful solo, unassisted, and unsupported two month trek across Antarctica. The primary purpose of this case history study was to acquire weekly DBS samples and analyze them for shifts in proteins during normal training and overtraining periods to discover additional biomarkers that could be used for NFOR and OTS detection. As in prior studies, participant burden was reduced by using DBS samples collected from fingerprick blood drops. The participant was trained in this technique to allow freedom of travel and improved compliance to the sampling regimen (weekly, Thursday mornings, overnight fasted).

## 2. Materials and Methods

### 2.1. Participant

The study participant for this case history study (33 years of age) was a professional endurance athlete, mountain climber, and adventurer. The participant voluntarily signed the informed consent form, and study procedures were submitted to and approved by the Institutional Review Board at Appalachian State University.

### 2.2. 28-Week Data Collection

Study procedures were developed to induce a low participant burden and were reviewed with the study participant prior to providing voluntary consent. Basic demographic, lifestyle, exercise training, and nutrient intake data were obtained using standard questionnaires and logs.

Physical fitness performance test scores were measured during three sessions at the Human Performance Lab, and included body composition, mean and peak anaerobic power during the Wingate 30-s sprint cycling test, and strength through handgrip and leg/back dynamometer tests. These lab fitness test sessions took place at the beginning of the 28-week monitoring period, and then after 7 weeks and 25 weeks (2 weeks post-Antarctica trek). Height and body weight were measured using a seca stadiometer and scale (Hanover, MD, USA). Body composition was measured with the Bod Pod body composition analyzer (Life Measurement, Concord, CA, USA). Leg-back dynamometer strength was assessed during a maximal lift test with arms straight and knees slightly bent using a bar that was attached to a platform via a chain and dynamometer (Lafayette Instruments, Lafayette, IN, USA). The test was repeated three times, with the highest score recorded. Handgrip dynamometer strength (Lafayette Instruments, Lafayette, IN, USA) was assessed using the best score from three maximal 2 to 3 s grips. The Lode cycle ergometer (Lode B.V., Groningen, The Netherlands) was used for the 30 s Wingate anaerobic power cycling test. The cycle ergometer was adjusted to the body mass of the subject (7 W per kilogram), with peak and total wattage power output recorded and adjusted to body mass.

Dried blood spot (DBS) specimens were collected via fingerprick onto standard blood spot cards (Whatman^®^ protein saver cards, Sigma-Aldrich, St. Louis, MO, USA). The participant was instructed in the fingerprick blood sample procedure, and samples were collected weekly in an overnight fasted state (26 July 2018 to 7 February 2019, Thursday mornings). The DBS samples were shipped to ProteiQ Biosciences (Berlin, Germany) for proteomics analysis.

Training distress was monitored weekly (Thursday mornings when acquiring the DBS samples) using the training distress scale (TDS). The TDS is a 19 item self-reported questionnaire that calculates training distress and performance readiness during the previous 48 h [22]. The participant responded to these items by indicating the extent to which the symptom was experienced using a 5-point bipolar scale anchored by the phrases not at all (0) and extreme amount (4).

Food and nutrient intake were monitored during the Antarctica trek (Weeks 16 to 23) using dietary recall. Food items were meticulously portioned out for each day prior to the trek. The recall consisted of a description of the food or beverage consumed with the brand name, quantity, and amount consumed, and the method of preparation. Nutrient analysis was conducted using the Food Processor software system (v. 11.7.1) (ESHA Research, Salem, OR, USA).

### 2.3. Proteomics Procedures

Punches, 4 mm diameter wide, were punched out of the DBS samples and proteins were resolubilized in 6 M urea, 50 mM ammonium bicarbonate (AmBiC) and 0.1 mM dithiothreitol for 30 min at 37 °C while shaking. Proteins were then alkylated by adding 0.1 mM iodoacetamide for 30 min in the dark at room temperature. After protein quantitation, 25 μg of proteins were taken for further processing. Proteins were diluted to a final volume of 50 μL with 50 mM AmBiC containing 1:50 ratio trypsin (Promega, V5111, Madison, WI, USA). Proteins were digested for 3 h at 37 °C while shaking. Digestion was quenched by adding 1% formic acid (FA). Subsequently, peptides were cleaned using C18 96-well plates (Waters Corporation, Milford, MA, USA). Peptides were eluted from the column with 50% acetonitrile (ACN)/0.1% FA. Samples were then lyophilized prior to nano-electrospray ionization liquid chromatography tandem mass spectrometry (nanoLC-MS/MS).

Before injection on the nanoLC-MS/MS, peptides were resolubilized in 50 μL of 0.1% FA, and 3% can and 1 μg of protein was used for nanoLC-MS/MS measurement. Pooled samples were run to monitor CV values and assess the quality of label-free quantitation. All samples were measured with a combination of a nano Acquity ultraperformance liquid chromatography (UPLC) system (Waters Corporation, Milford, MA, USA) and a Thermo Scientific Orbitrap Fusion Tribrid Mass Spectrometer in data independent analysis (DIA) mode (Thermo Fisher Scientific, Waltham, MA, USA). Peptides were separated on an analytical column ethylene bridged hybrid (BEH) C18, 130A, 1.7 μm, 75 μm × 150 mm (Waters Corporation, Milford, MA, USA). Flowrate was 300 nL/min (buffer A, HPLC H_2_O, 0.1% formic acid; buffer B, ACN, 0.1% formic acid; 60 min run method, 40 min gradient 0 to 3 min 2% buffer B, 3 to 40 min nonlinear stepwise gradient from 2% to >40% buffer B, 40 to 45 min 95% buffer B, 45 to 60 min 2% buffer B). Eighteen MS2 windows were used for DIA with different widths for equal distribution of MS1 precursor intensity and a cycle time that led to ~10 points per chromatographic peak. After each cycle, 1 MS1 acquisition was inserted. MS2 was done at 30,000 resolution and an MS filling time of 54 ms. The automatic gain control (AGC) target was set on 2.0 × 10^5^. The MS1 resolution was 60,000 with an MS filling time of 110 ms and an AGC target of 1.0 × 10^6^. An in-house dried blood spot library was used that was created previously.

### 2.4. Data Processing

The DIA files were processed by infineQ software (www.infineq.com) that is based on the DIA-neural networks algorithm [23] using standard settings with a false discovery rate (FDR) cut-off set to 1% for precursors. Retention times of the library were adapted to instrument specific retention times. Data was normalized using the CyCloess normalization approach and finally peptides were quantified into proteins.

### 2.5. Statistics

The 28-week period was divided into the following time segments based on the TDS scores: Weeks 1 to 4 (recovery and training following the 50 state peaks challenge event), Weeks 5 to 8 (relaxed Greenland training and practice), Weeks 9 to 15 (physical training, with an emphasis on strength), Weeks 16 to 19 (first month of Antarctica trek), Weeks 20 to 23 (second month of Antarctica trek), and Weeks 24 to 28 (recovery period). Weeks 5 to 8 had the lowest TDS scores and this time segment was set as the comparison period for all other time segments, with a focus on contrasts with the last portion of the Antarctica trek (Weeks 20 to 23) when TDS scores were highest.

For the linear models, the time segments comparisons were made by calculating the log2 fold-change ratios with the limma R package (v. 3.6.2, Bioconductor, Buffalo, NY, USA) [24] using as reference Weeks 5 to 8 (the Greenland period). The log2 ratios were obtained by defining a linear model for each protein. This approach estimates the mean values per time segment, and for each comparison between means, a two-sample moderated t-test is used to infer a p-value. The reported *p*-values were corrected for false discovery rate (FDR) and tested against an alpha value < 0.1.

For the supervised classification analysis, we used sPLS-DA (sparse partial least squares discriminant analysis) [25] and LASSO (least absolute shrinkage and selection operation) [26] classification algorithms to find biomarkers specific to each time segment in order to discriminate among the different time segments. LASSO is a powerful regularization technique and incorporates an L1- penalization term into the loss function forcing some coefficients to be zero. Differences between scores from LASSO output were compared using the Kruskal–Wallis nonparametric rank sum test.

### 2.6. Protein–Protein Interaction Network Analysis

Proteins expressed were mapped onto STRING v11 to build protein–protein interaction (PPI) networks. STRING v11 (search tool for the retrieval of interacting genes and proteins) is a database of known and predicted physical and functional protein associations based on genomic context, high-throughput experiments, co-expression, and previous knowledge (http://string-db.org/) [5].

## 3. Results

The successful Antarctica trek covered 1500 km over 54 days from 3 November to 26 December 2018. The total TDS score was lowest during the four weeks of training in Greenland and peaked during the last four weeks of the Antarctica trek (Figure 1).

Physical fitness tests were conducted at Weeks 1, 9, and 25 (two weeks after completing the Antarctica trek) (Figure 2). The participant experienced a decrease of 11.4 kg in body mass, a 26% decrease in leg/back lifting strength, and an 18% and 30% decrease in anaerobic mean and peak power, respectively.

During the Antarctica trek, the participant consumed an average of 7048 kcal/day, with a macronutrient intake energy distribution of 45% carbohydrate, 44% fat, and 13% protein. This included approximately 4138 kcal/day from nutrient-dense sports bars (Standard Process Inc., Palmyra, WI, USA).

A total of 712 proteins were identified from the proteomics procedures (Appendix A). When the identified peptides were not clearly linked to a specific isoform (e.g., histones), the isoforms were classified as one protein. Supervised classification using sPLS-DA and LASSO successfully separated Weeks 20 to 23 (second month of the Antarctica trek) from all other week segments. The contrast analysis between Weeks 5 to 8 (Greenland training) and Weeks 20 to 23 showed that 31 proteins were upregulated (Table 1) and 35 (Table 2) were downregulated.

Protein–protein interaction (PPI) networks were constructed separately for the proteins listed in Table 1 and Table 2 using the Search Tool for Retrieval of Interacting Genes/Proteins (STRING). Of the 31 proteins listed in Table 1 that increased during the last four weeks of the Antarctica trek, *n* = 20 were included in immune system-related, biological process GO terms, with an average local cluster coefficient of 0.697 (PPI enrichment *p*-value < 0.0001) (Figure 3). The mean log-fold increase for all 20 proteins was 0.85 during the last four weeks of the Antarctica trek and 0.92 during the five weeks of recovery as compared with the reference week segment (Weeks 5 to 8, Greenland training). Biological process GO terms from STRING supported an increase in regulation of the immune system process, complement activation, proteolysis, the inflammatory response, platelet degranulation, and leukocyte mediated immunity. Most of the upregulated proteins were extracellular or secreted.

During the Antarctica trek, the participant consumed approximately 8000 kilocalories per day, and this was supplied from high-fat energy bars that were formulated for this event. Fourteen proteins from Table 1 were included in nutrition-related, biological process GO terms, with an average local cluster coefficient of 0.883 (PPI enrichment p-value < 0.0001) (Figure 4). The mean log-fold increase for all 14 proteins was 1.00 during the last four weeks of the Antarctica trek and 0.99 during the five weeks of recovery as compared with the reference week segment (Weeks 5 to 8, Greenland training). Biological process GO terms from STRING supported an increase in plasma lipoprotein particle remodeling, regulation of lipid transport, retinoid metabolic process, and vitamin transport. Most of the downregulated proteins were intracellular.

Of the 35 proteins listed in Table 2 that decreased during the last four weeks of the Antarctica trek, *n* = 17 were included in immune system-related, biological process GO terms, with an average local cluster coefficient of 0.741 (PPI enrichment p-value <0.0001) (Figure 5). The mean log-fold decrease for all 17 proteins was −0.83 during the last four weeks of the Antarctica trek and −0.33 during the five weeks of recovery as compared with the reference week segment (Weeks 5 to 8, Greenland training). Biological process GO terms from STRING supported a decrease in the immune system process, neutrophil degranulation, vesicle mediated transport, and antimicrobial humoral response.

During the 5-week recovery period, three DBS samples were obtained (Weeks 1, 4, and 5). A total of 65 proteins (42 immune-related) were significantly upregulated as compared with Weeks 5 to 8, and 65 (30 immune-related) were significantly downregulated (Appendix A). Of the upregulated immune-related proteins, most were linked to inflammation with some of the greatest fold increases seen for attractin, serum amyloid A4, serpin family F member 1, kallikrein B1, fibronectin 1, alpha 2-HS glycoprotein, inter-alpha-trypsin inhibitor heavy chain4, vitronectin, hemopexin, kininogen 1, protein S, and several complement proteins (Appendix A). Of the downregulated immune-related proteins, seven were related to the IL-12-mediated signaling pathway, 10 to neutrophil degranulation, and 13 to actin filament and cytoskeleton organization (Appendix A).

## 4. Discussion

An elite adventure athlete was successfully monitored for 28 weeks, with weekly DBS samples analyzed for shifts in 712 blood proteins. The athlete cycled through various phases of training that culminated in the successful solo trek across the Antarctica. Protein data from the DBS samples were grouped according to training periods based on TDS scores, with the greatest contrast seen between Weeks 5 to 8 (relaxed Greenland training) and the last month of the Antarctica trek (Weeks 20 to 23) when shifts in 67 blood proteins were observed. Of these proteins, 31 were upregulated, and 35 were downregulated, with over half (56%) related to immune system function and 14 linked to nutrition-related processes.

The log-fold change in upregulated immune-related proteins (*n* = 20) was considerable and was maintained throughout five weeks of recovery from the Antarctica trek. The PPI analysis and related GO terms supported an increase in regulation of the immune system process, especially leukocyte mediated immunity, complement activation, the inflammatory response, and platelet degranulation. At the same time, 14 proteins linked to nutritional effects were upregulated. PPI and GO terms supported an increase in plasma lipoprotein particle remodeling, regulation of lipid transport, retinoid metabolic process, and vitamin transport. Eight of these proteins performed dual roles with key involvement in the immune response, and these included apolipoproteins (apo) A1, A2, D, and E, retinol binding protein, clusterin, transthyretin, and angiotensinogen.

ApoE interacts with the low-density lipoprotein receptor (LDLR) to mediate the transport of cholesterol- and triglyceride-rich lipoprotein particles into cells via receptor-mediated endocytosis [27]. In contrast, apoAI is the major protein constituent of high-density lipoprotein (HDL) that mediates reverse cholesterol transport out of cells. ApoE and apoA1 are synthesized primarily in the liver but can also be expressed by lung cells where they help attenuate inflammation, oxidative stress, and tissue remodeling responses, while augmenting adaptive immunity and host defense [27]. ApoE receptor 2, one of the LDLR family members expressed in macrophages, can bind to its ligand apoE, exhibiting an anti-inflammatory role in atherosclerosis [27]. Additional evidence suggests that apoE and C-reactive protein (CRP) are negatively related [28]. Our data support that apoA1, apoA2, apoD, and apoE are involved in lipid transport, but also play a role in the immune and inflammation response to stressful levels of exercise. This appears to be a novel finding that will require additional research.

Recent findings support that clusterin is also involved in both lipid transport and inflammation [29]. Clusterin, also known as apolipoprotein J, is induced in response to a wide variety of tissue injuries. Clusterin has chaperone activity, is a functional homolog to small heat shock proteins, and binds hydrophobic domains of numerous non-native proteins, targeting them for receptor-mediated internalization and lysosomal degradation. Clusterin also interacts with a broad spectrum of molecules including lipids, components of the complement system, amyloid-forming proteins, and immunoglobulins [30]. Our data support that clusterin played a dual role in regulating the immune response during the stressful Antarctica trek and in interacting with lipoprotein particle remodeling. This finding has not been previously reported.

Upregulated immune-related proteins during the Antarctica trek and five weeks of recovery included numerous proteins from the complement system. The complement system is composed of over 30 proteins and becomes activated in response to overreaching during athletic training, tissue injury, invading pathogens, or exposure to other foreign surfaces [4,7,31,32]. Complement 5a (C5a), for example, is secreted by liver cells and macrophages, and is essential to the innate immune response, and promotes inflammatory reactions. C5a is an important proinflammatory mediator that is cleaved enzymatically from C5 on activation of the complement cascade. C5a is quickly metabolized by carboxypeptidases (as supported by our data in Figure 3), forming the less-potent C5a des arginine (desArg). C5a and C5a desArg interact with their receptors resulting in widespread effects essential to the immune response including clearance of pathogens, host defense, increased vascular permeability, chemotaxis of inflammatory cells, respiratory burst activity, cytokine and chemokine release, phagocytosis, adaptive immunity, and coagulation [32]. Our data support a large increase in C5a and complement activation in response to extended mental and physical stress and is consistent with prior proteomics-based studies from our research group [4,7].

Lipocalin proteins are involved in inflammation caused by immune system activation. Lipocalins include several proteins that were upregulated during the Antarctica trek including apoD, retinol-binding protein, and C8 gamma (C8G) [33,34]. ApoD is an acid glycoprotein and is elevated in disease states such as prostate cancer and Alzheimer’s disease. C8G is a part of the complement membrane attack complex, and we have previously shown that this complement is elevated during overreaching [4,7]. After secretion, retinol binding protein complexes with another plasma protein, transthyretin, which is a triiodothyronine binding protein in humans. Retinal binding protein is linked to other inflammatory markers and can induce the secretion of cytokines and adhesion molecules in macrophages and endothelial cells [34].

Transcortin, also known as corticosteroid-binding globulin (CBG) or serpin A6, is a protein produced in the liver in animals. In humans it is encoded by the SERPINA6 gene and an alpha globulin [35,36]. CBG was increased during the Antarctica trek, and has evolved as an important biomarker for exercise overreaching and overtraining [4,7]. The importance of CBG is highlighted by its ability to bind 80% to 90% of cortisol in plasma, leaving only about 4% to 5% circulating in the free fraction and the remainder bound loosely to albumin. CBG plays a role in the control of the inflammatory response, gluconeogenesis, and stress. While CBG does not act as a protease inhibitor, it is a substrate for neutrophil elastase [36].

Kallistatin is a unique serine protease inhibitor (serpin family A member 4) and was one of several serpins elevated during the Antarctica trek and five weeks of recovery. Kallistatin has many roles including suppression of cytokine signaling expression in macrophages [37,38]. Kallistatin antagonizes tumor necrosis factor (TNF)-α induced inflammation, oxidative stress, and apoptosis while enhancing bacterial clearance and exerting anti-inflammatory effects. The elevation of kallistatin during the Antarctica trek and recovery appears to represent one attempt by the immune system to restore homeostasis [38].

Another elevated protein was plasminogen, an acute phase protein which is the zymogen form of the serine protease plasmin [39]. Plasminogen plays a crucial role in fibrinolysis, as well as wound healing, immunity, tissue remodeling, and inflammation. Cellular uptake of fibrin degradation products leads to apoptosis, which represents one of the pathways for crosstalk between fibrinolysis and tissue remodeling. Plasminogen was one of many acute phase proteins that were significantly elevated either during the last month of the Antarctica trek or during recovery, and these included complements (C3, C4, factor B, CI inhibitor, or serpin family G member 1), protein S, vitronectin, inter-α-trypsin inhibitor, hemopexin, serum amyloid A, fibronectin, angiotensinogen, transthyretin, and α-2 HS glycoprotein. We and others have shown that acute phase proteins are important biomarkers for intense exertion and overreaching, and influence one or more stages of inflammation [4,7,14,15,16,17,18].

The upregulated of 20 immune-related proteins was countered by a downregulation of 16 proteins linked to a decrease in the immune system process, especially neutrophil degranulation, vesicle mediated transport, and antimicrobial humoral response.

Neutrophils are the most abundant leukocytes in the circulation, and recruitment and activation of these cells are crucial for defense against invading pathogens [40,41]. Neutrophils respond quickly and deploy cytosolic granules containing enzymatic and chemical effectors. Azurophilic granules, specific granules, gelatinase granules, and secretory vesicles in neutrophils each have specific types of proteins and effectors that are released depending on the signaling pathway, context, and function outcome [41]. Azurophilic granules, for example, contain oxidant-producing enzymes such as myeloperoxidase, proteases such as elastase and cathepsin G, and membrane-permeabilizing proteins such as lysozyme and defensins. Inappropriate recruitment and activation of neutrophils can lead to tissue damage during an exaggerated inflammatory response. Neutrophil degranulation is tightly regulated through a multistep process involving calcium-dependent and kinase-dependent signaling pathways, actin and microtubule reorganization pathways mediated by calcium and nucleotide guanosine triphosphase hydrolase enzymes, and fusion process cell-surface receptors [41]. Thus, precise control of neutrophil movement and degranulation is of particular importance and can explain why neutrophil degranulation was suppressed during the most stressful phase of the Antarctica trek. This viewpoint is supported by the strong decrease in blood S100-A8/A9 (calprotectin) during the last month of the Antarctica trek. Calprotectin, the most abundant protein in the neutrophil, is released during trauma, stress, and infection, promotes phagocyte migration and inflammation, and functions as an alarmin and endogenous danger-associated molecular pattern (DAMP) [42,43]. Excessive expression of calprotectin magnifies the inflammatory process and related damage, induces the secretion of multiple cytokines in inflammatory cells, and if not properly regulated, can induce a vicious cycle in certain disorders [43]. Together, these data suggest that neutrophil function and degranulation were strongly moderated when inflammation was high due to stressful exercise levels.

Galectin-3 promotes fibroblast proliferation and transformation and stimulates the phagocytosis of apoptotic cells and cellular debris by macrophages. Galectin-3 is highly expressed and secreted by macrophages [44,45]. Interleukin (IL)-10 increases the expression of intracellular galectin-3 through activation of signal transducer and activator of transcription 3 (STAT3) [44]. The low levels of galectin-3 and IL-10 during the physiologically stressful Antarctica trek is another indicator of diminished immune function. During short-term overreaching, galectin-3 is elevated [4,7], but the data from the current study indicates that downregulation occurs during extended training distress.

Several proteins that were downregulated during the Antarctica trek were related to actin cytoskeleton organization. Calprotectin, which was strongly decreased during the trek, plays a significant role in mediating the rapid rearrangement of the cytoskeleton, a prerequisite for successful cell migration, phagocytosis, and exocytosis [43]. The actin cytoskeleton is a complex network controlled by an array of actin-binding proteins including plastins that non-covalently crosslink actin filaments into tight bundles [46]. Elongation factors, which were also downregulated during the trek, are essential for protein synthesis and have multiple immune-related roles including promotion of actin and cytoskeleton organization, detection and targeting of misfolded proteins for proteolytic degradation, and induction of cytotoxic T cells and heat shock protein 70 [47]. Cytoplasmic dynein 1 acts as a motor for the intracellular retrograde motility of vesicles and organelles along microtubules, is involved with neutrophil degranulation, and can help coordinate actin and microtubule organization at the immune synapse [48]. F-actin-capping protein subunit alpha-1 regulates growth of the actin filament by capping the barbed end (plus-end) of growing actin filaments, preventing any further assembly from occurring. Moesin is a major component of the cytoskeleton in neutrophils and helps link filamentous actin to the plasma membrane [49]. Moesin contributes to the slow rolling and subsequent recruitment of neutrophils during inflammation [50]. Our data support a decrease in moesin, especially during the 5-week recovery time period. Stomatin, another downregulated protein during the five weeks of recovery, can have a structural role for the anchorage to the actin cytoskeleton in neutrophils [51]. These data collectively support that actin cytoskeleton remodeling was decreased during the Antarctica trek and recovery, adding to the overall finding that neutrophil function and degranulation were mitigated, perhaps to reduce undue tissue damage.

## 5. Conclusions

Recent improvements in MS-based proteomics procedures allow highly specific measurements of multiple protein patterns from small amounts of blood [52]. This greatly improves upon the earlier pursuit of NFOR and OTS biomarkers that used a few targeted outcomes, and offers a fresh, unbiased, hypothesis-free approach. In this study, proteomics monitoring of an elite adventure athlete during a 28-week period of normal training and overtraining revealed up- and downregulation of 14 nutrition-related and 37 immune-related proteins. These protein shifts were most evident when energy expenditure was highest, body mass was reduced, and training distress was most severe during the last month of the trek across the Antarctica. The athlete also experienced a decrease in performance measures that is consistent with NFOR and OTS [1]. These case history data could or coud not be applicable to other athletes but will provide direction for future studies. The DBS proteomics procedure is not able to distinguish proteins derived from intracellular and extracellular sources. Nonetheless, using available information, we determined that most of the upregulated proteins were from extracellular sources or were secreted, and that most of the downregulated proteins were from intracellular sources.

There is scant evidence available regarding most of these immune-related proteins within the field of exercise and nutrition immunology, in part due to an underutilization of proteomics methods. The PPI analysis and related GO terms supported an increase in regulation of the immune system process highlighted by inflammation, complement activation, and platelet degranulation that occurred at the same time that neutrophil degranulation, vesicle mediated transport, and antimicrobial humoral responses were suppressed. The heightened immune response continued unabated during the 5-week recovery process. Many of the upregulated and downregulated immune-related proteins identified in this case history study can be regarded as candidate biomarkers for NFOR and OTS in future studies of athletic groups. On the basis of our prior studies [4,7] and the data from this study, key NFOR and OTS biomarkers would include the following upregulated immune-related proteins (*n* = 28 during both the Antarctica trek and recovery): kallistatin (serpin family A member 4), plasma protease C1 inhibitor (serpin family G member 1), complement proteins (C1r, C1s, C2, C3, C4-A, C5, C8 gamma, factor I, factor B, factor H), carboxypeptidase N subunits 1 and 2, angiotensinogen, inter-alpha-trypsin inhibitor heavy chain 4, plasminogen, corticosteroid-binding globulin (serpin family A member 6), alpha-1-B glycoprotein, clusterin, attractin, serpin family F member 1 (pigment epithelium-derived factor), plasma kallikrein, fibronectin, alpha 2-HS glycoprotein, vitronectin, kininogen-1, and serum amyloid A-4. Nine downregulated proteins are also included on the NFOR and OTS list which include: proliferation-associated 2G4, IL-10, galactin-3, calcineurin like phosphoesterase domain containing 1, dynein cytoplasmic 1 heavy chain 1, S100A8 and S100A9, moesin, and stomatin. Taken together, these proteomics data support a dichotomous immune response to sustained physiological stress in the harsh environment of the Antarctica highlighted by inflammation and complement activation with downregulated neutrophil degranulation and humoral immunity.

## Figures and Tables

**Figure 1 proteomes-08-00004-f001:**
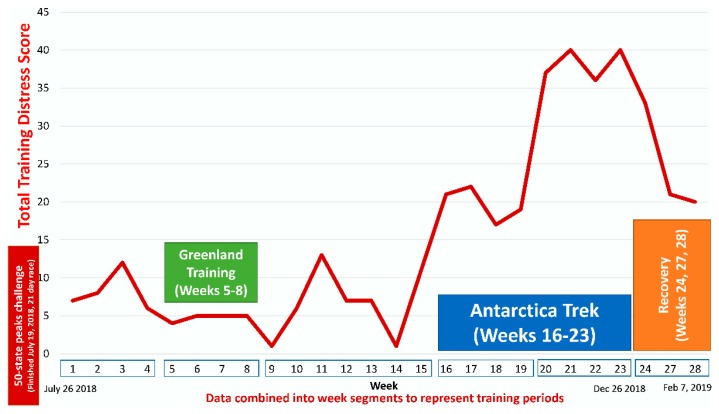
The total training distress score (TDS) during the 28-week monitoring period.

**Figure 2 proteomes-08-00004-f002:**
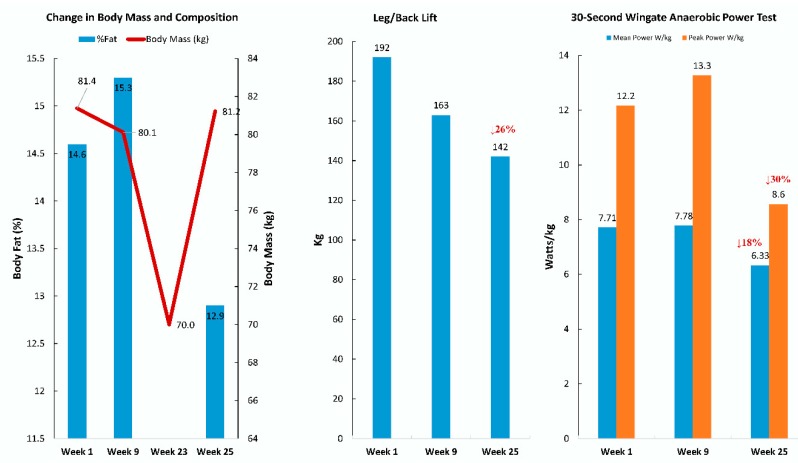
Physical fitness test scores at Weeks 1, 9, and 25.

**Figure 3 proteomes-08-00004-f003:**
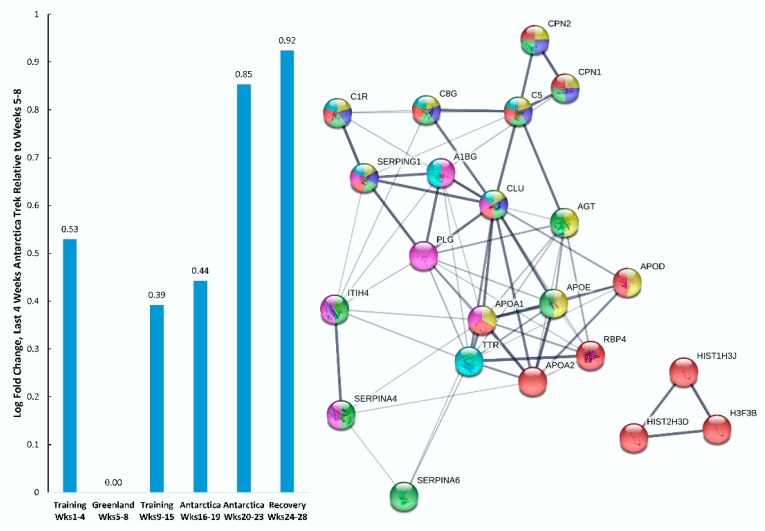
Protein–protein interaction (PPI) network for blood immune-related proteins (*n* = 22) that increased during the last four weeks of the Antarctica trek as compared with the reference week segment (Weeks 5 to 8). Gene ontology (GO) terms for biological process were coded as follows: Red, regulation of immune system process; dark blue, regulation of complement activation; green, regulation of proteolysis; yellow, regulation of inflammatory response; pink, platelet degranulation; and light blue, leukocyte mediated immunity. Acronyms represent upregulated genes (see Table 1 for descriptions).

**Figure 4 proteomes-08-00004-f004:**
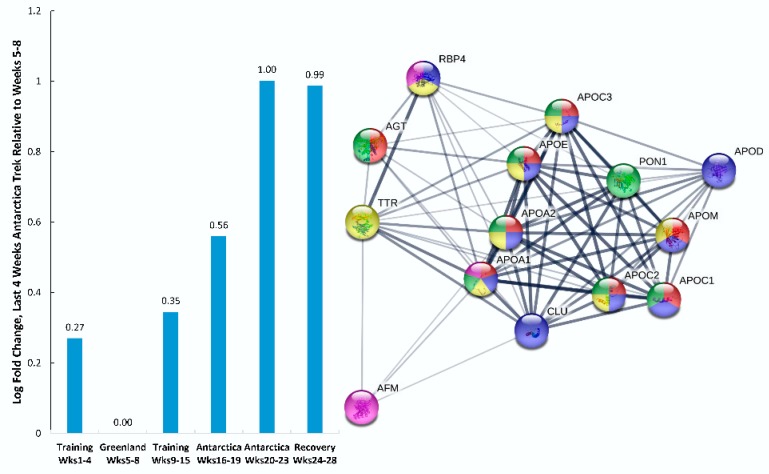
PPI network for nutrition-related proteins (*n* = 14) that increased during the last four weeks of the Antarctica trek as compared with the reference week segment (Weeks 5 to 8). GO terms for biological process were coded as follows: Red, plasma lipoprotein particle remodeling; dark blue, lipid transport; green, regulation of lipid transport; yellow, retinoid metabolic process; and pink, vitamin transport. Acronyms represent upregulated genes (see Table 1 for descriptions).

**Figure 5 proteomes-08-00004-f005:**
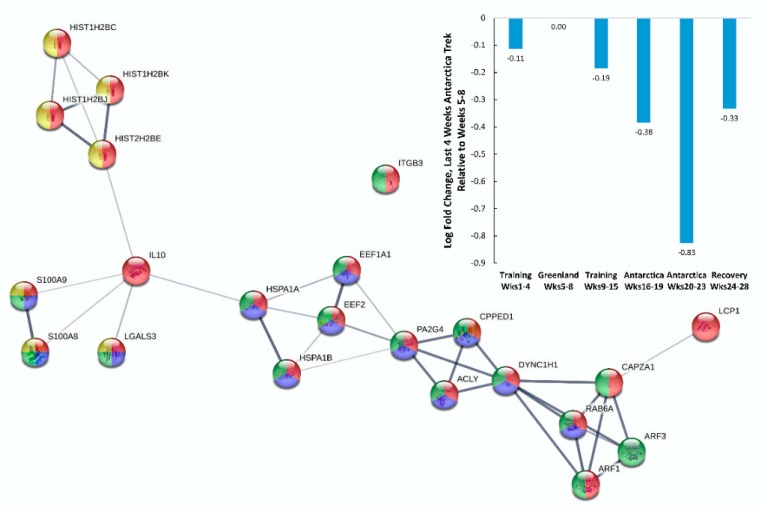
PPI network for immune-related proteins (*n* = 17) that decreased during the last four weeks of the Antarctica trek as compared with the reference week segment (Weeks 5 to 8). GO terms for biological process were coded as follows: Red, immune system process; dark blue, neutrophil degranulation; green, vesicle mediated transport; and yellow, antimicrobial humoral response. Acronyms represent downregulated genes (see Table 2 for descriptions).

**Table 1 proteomes-08-00004-t001:** Proteins upregulated (*n* = 31) during the last month (Weeks 20–23) of the Antarctica trek as compared with Weeks 5 to 8 (Greenland training) (adjusted *p*-value < 0.100). Proteins are ordered by log-fold increase. Proteins in bold were involved with the immune response in protein–protein interaction (PPI) networks. *^†^* = extracellular or secreted; *^‡^* = intracellular.

Gene	UniProt Identifier	Protein Description	Log-Fold Change	*p*-Value	Adjusted *p*-Value
**APOE**	**P02649**	**apolipoprotein E** *^†^*	1.856	0.000	0.000
GPX3	P22352	glutathione peroxidase 3*^†^*	1.590	0.000	0.000
APOC3	P02656	apolipoprotein C3*^†^*	1.574	0.000	0.001
**APOD**	**P05090**	**apolipoprotein D** *^†^*	1.559	0.000	0.000
HGFAC	Q04756	HGF activator*^†^*	1.467	0.000	0.006
APOC2	P02655	apolipoprotein C2*^†^*	1.463	0.000	0.003
**RBP4**	**P02753**	**retinol binding protein 4** *^†^*	1.412	0.000	0.001
APOM	O95445	apolipoprotein M*^†^*	0.984	0.000	0.007
HDHD2	Q9H0R4	haloacid dehalogenase like hydrolase domain containing 2*^†^*	0.976	0.001	0.038
**SERPINA4**	**P29622**	**serpin family A member 4; kallistatin** *^†^*	0.971	0.002	0.038
APOC1	P02654	apolipoprotein C1*^†^*	0.970	0.000	0.003
**SERPING1**	**P05155**	**serpin family G member 1; plasma protease C1 inhibitor** *^†^*	0.947	0.002	0.041
CLNS1A	P54105	chloride nucleotide-sensitive channel 1A*^‡^*	0.898	0.007	0.076
**HIST1H3A–J**; H3F3A; **H3F3B**; HIST3H3; **HIST2H3A,C,D**	**P68431;** P84243; Q16695; Q71DI3	**histone cluster 1 H3 family member a–j;***^‡^* **H3 histone family members 3A,3B; ***^‡^* histone cluster 3 H3; *^‡^* **histone cluster 2 H3 family member a,c,d***^‡^*	0.897	0.006	0.074
**C8G**	**P07360**	**complement C8 gamma chain** *^†^*	0.864	0.008	0.086
**C1R**	**P00736**	**complement C1r** *^†^*	0.810	0.001	0.028
**APOA1**	**P02647**	**apolipoprotein A1** *^†^*	0.797	0.000	0.001
**CPN2**	**P22792**	**carboxypeptidase N subunit 2** *^†^*	0.793	0.002	0.038
**CPN1**	**P15169**	**carboxypeptidase N subunit 1** *^†^*	0.763	0.002	0.041
**AGT**	**P01019**	**angiotensinogen** *^†^*	0.680	0.002	0.041
**ITIH4**	**Q14624**	**inter-alpha-trypsin inhibitor heavy chain 4** *^†^*	0.648	0.001	0.033
**PLG**	**P00747**	**plasminogen** *^†^*	0.621	0.000	0.017
**APOA2**	**P02652**	**apolipoprotein A2** *^†^*	0.603	0.000	0.019
AFM	P43652	afamin*^†^*	0.602	0.004	0.064
PON1	P27169	paraoxonase 1*^†^*	0.590	0.002	0.042
**SERPINA6**	**P08185**	**serpin family A member 6 (cortisol binding globulin) ** *^†^*	0.585	0.001	0.034
**A1BG**	**P04217**	**alpha-1-B glycoprotein** *^†^*	0.577	0.001	0.038
**C5**	**P01031**	**complement C5** *^†^*	0.547	0.003	0.054
**TTR**	**P02766**	**transthyretin** *^†^*	0.498	0.004	0.064
AZGP1	P25311	alpha-2-glycoprotein 1, zinc-binding*^†^*	0.468	0.006	0.075
**CLU**	**P10909**	**clusterin** ^*† ‡*^	0.439	0.002	0.038

**Table 2 proteomes-08-00004-t002:** Proteins downregulated (*n* = 35) during the last month (Weeks 20 to 23) of the Antarctica trek compared to Weeks 5 to 8 (Greenland training) (adjusted *p*-value < 0.100). Proteins are ordered by log -old decrease. Proteins in bold were involved with the immune response in protein–protein interaction (PPI) networks. *†* = extracellular or secreted; *^‡^* = intracellular.

Gene	UniProt Identifier	Protein Description	Log-Fold Change	*p*-Value	Adjusted *p*-Value
**ACLY**	**P53396**	**ATP citrate lyase** *^‡^*	−0.274	0.008	0.088
YWHAE	P62258	tyrosine 3-monooxygenase/tryptophan 5-monooxygenase activation protein epsilon*^‡^*	−0.287	0.004	0.065
ARF1; ARF3	P61204; P84077	ADP ribosylation factors 1,3*^‡^*	−0.300	0.005	0.065
PSMC6	P62333	proteasome 26S subunit, ATPase 6*^‡^*	−0.319	0.006	0.074
PSMD9	O00233	proteasome 26S subunit, non-ATPase 9*^‡^*	−0.356	0.008	0.089
PSMC5	P62195	proteasome 26S subunit, ATPase 5*^‡^*	−0.358	0.009	0.095
**PA2G4**	**Q9UQ80**	**proliferation-associated 2G4** *^‡^*	−0.383	0.002	0.041
**HSPA1A; HSPA1B**	**P08107**	**heat shock protein family A (Hsp70) members 1A, 1B** *^‡^*	−0.386	0.003	0.045
XPO7	Q9UIA9	exportin 7*^‡^*	−0.392	0.006	0.076
**IL10**	**P22301**	**interleukin 10** *^†^*	−0.408	0.006	0.074
**CAPZA1**	**P52907**	**capping actin protein of muscle Z-line subunit alpha 1** *^†^*	−0.432	0.000	0.017
IGLV1-47	P01700	immunoglobulin lambda variable 1-47*^†^*	−0.447	0.005	0.065
IGKC	P01834	immunoglobulin kappa constant*^†^*	−0.472	0.002	0.044
**LGALS3**	**P17931**	**galectin 3** *^†‡^*	−0.479	0.007	0.078
**LCP1**	**P13796**	**lymphocyte cytosolic protein 1** *^‡^*	−0.480	0.002	0.043
CALM1,2,3	P62158	calmodulin 1,2,3*^‡^*	−0.484	0.002	0.038
PCBP1	Q15365	poly(rC) binding protein 1*^‡^*	−0.544	0.006	0.074
AHSP	Q9NZD4	alpha hemoglobin stabilizing protein*^†‡^*	−0.607	0.001	0.033
RGS10	O43665-3	regulator of G protein signaling 10*^‡^*	−0.744	0.003	0.051
**CPPED1**	**Q9BRF8**	**calcineurin like phosphoesterase domain containing 1** *^†‡^*	−0.773	0.003	0.052
DHRS11	Q6UWP2	dehydrogenase/reductase 11*^† ‡^*	−0.794	0.005	0.071
ADK	P55263	adenosine kinase*^‡^*	−0.837	0.002	0.043
STMN2	Q93045	stathmin 2*^‡^*	−0.860	0.001	0.024
FSCB	Q5H9T9	fibrous sheath CABYR binding protein*^†‡^*	−0.891	0.004	0.063
**EEF2**	**P13639**	**eukaryotic translation elongation factor 2** *^‡^*	−0.897	0.000	0.006
YWHAB; YWHAG; YWHAQ	P31946; P61981; P27348	tyrosine 3-monooxygenase/tryptophan 5-monooxygenase activation protein beta; gamma; theta*^‡^*	−0.897	0.006	0.074
**ITGB3**	**P05106**	**integrin subunit beta 3** *^‡^*	−1.081	0.004	0.059
**RAB6A**	**P20340-2**	**RAB6A, member RAS oncogene family** *^‡^*	−1.090	0.003	0.053
**EEF1A1**	**P68104;** **Q5VTE0**	**eukaryotic translation elongation factor 1 alpha 1** *^‡^*	−1.190	0.001	0.033
HBM	Q6B0K9	hemoglobin subunit mu*^‡^*	−1.196	0.000	0.001
**HIST1H2BB-K, M–O**	P33778;P62807; P58876;Q93079; P06899;O60814; Q99879;Q99877; P23527;Q16778; Q5QNW6	**histone cluster 1 H2B family members b–k, m–o; ** *^‡^* **cluster 2 H2B family members e,f** *^‡^*	−1.301	0.000	0.020
CLLU1OS	Q5K130	chronic lymphocytic leukemia upregulated 1 opposite strand*^†‡^*	−1.303	0.002	0.040
**DYNC1H1**	**Q14204**	**dynein cytoplasmic 1 heavy chain 1** *^‡^*	−1.325	0.000	0.018
**S100A9**	**P06702**	**S100 calcium binding protein A9** *^†^*	−1.357	0.006	0.075
**S100A8**	**P05109**	**S100 calcium binding protein A8** *^†^*	−1.680	0.001	0.032

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
