# Peer review of "Proteomics-Based Detection of Immune Dysfunction in an Elite Adventure Athlete Trekking Across the Antarctica"

_proteomes, 2020, doi:10.3390/proteomes8010004_

Round 1
Reviewer 1 Report
This study revealed dynamic change of blooad proteome in an elite adventure athlete during his trekking across Antarctica. Although the data is valuable, there are still several issues needed to be addressed.
Comment 1: Since dried blood spots cannot be used to differentiate proteins from plasma and from blood cells, it is hard to normalize protein level by traditional DIA approaches which usually use MS2 spectra as a base for normalization. Ideally, secretory proteins in plasma and intracellular proteins in blood cells should be normalized separately. The authors should add this point in the Discussion and try to suggest a better calculating method to address the normalization issue. Comment 2: Only one athlete took part this study. This is also a weak point of the study, as the proteome changes of one person cannot be regarded as a conclusion for all individuals. The authors should add this point in the discussion. Comment 3: Blood is a complicated sample with proteins from many sources, such as secretory proteins from liver, secretory proteins from blood cells and intracellular proteins from blood cells. The authors should note the sources and exiting spaces of proteins in the table for giving a clear view for readers. Comment 4: Secretory proteins can exist in both intracellular (before secretion) and extracellular (after secretion) spaces. They have different meanings. For examples, an inflammatory cytokine increase within the white blood cells means that the cells are ready to fight against microbes, while an increase of cytokine in plasma means that inflammation is undergoing. The author should let readers know that dried blood spots cannot provide such information, and the proteome changes of the two spaces (intracellular vs. extracellular) still need further study to clarify.Author Response
Comment 1: Since dried blood spots cannot be used to differentiate proteins from plasma and from blood cells, it is hard to normalize protein level by traditional DIA approaches which usually use MS2 spectra as a base for normalization. Ideally, secretory proteins in plasma and intracellular proteins in blood cells should be normalized separately. The authors should add this point in the Discussion and try to suggest a better calculating method to address the normalization issue. Comment 3: Blood is a complicated sample with proteins from many sources, such as secretory proteins from liver, secretory proteins from blood cells and intracellular proteins from blood cells. The authors should note the sources and exiting spaces of proteins in the table for giving a clear view for readers. Comment 4: Secretory proteins can exist in both intracellular (before secretion) and extracellular (after secretion) spaces. They have different meanings. For examples, an inflammatory cytokine increase within the white blood cells means that the cells are ready to fight against microbes, while an increase of cytokine in plasma means that inflammation is undergoing. The author should let readers know that dried blood spots cannot provide such information, and the proteome changes of the two spaces (intracellular vs. extracellular) still need further study to clarify.
RESPONSE: We are responding to comments 1,3,4 together.
DBS proteomics gives an overview of what is happening to blood protein levels after a defined stimulus. During the procedure, we punch a large middle section of the DBS, prepare the sample, and then identify proteins from within the complete mixture of blood that includes plasma proteins, blood cell proteins, and secreted proteins, without any fractionation. So yes, with the DBS proteomics approach, we are measuring all of the available proteins, and do not distinguish proteins from intracellular or extracellular sources. In response to your concerns, we added symbols to Tables 2 and 3 to indicate whether the protein came from an extracellular source or was secreted, or came from an intracellular source, based on available information. We also added statements in the results section: "Most of the upregulated proteins were extracellular or secreted." "Most of the downregulated proteins were intracellular." We added a limitation statement to the discussion indicating that the DBS proteomics procedure includes proteins from both intracellular and extracellular sources.
Comment 2: Only one athlete took part this study. This is also a weak point of the study, as the proteome changes of one person cannot be regarded as a conclusion for all individuals. The authors should add this point in the discussion.
RESPONSE: Added this statement to the conclusion section: "These case history data may or may not be applicable to other athletes, but will provide direction for future studies."
Reviewer 2 Report
I have carefully examined the manuscript entitled “Proteomics-Based Detection of Immune Dysfunction in an Elite Adventure Athlete Trekking Across Antarctica” by Nieman et al. Authors describe an efficient approach to measure changes in proteome of an athlete submitted to a hard mental and physical stress during a trek across Antarctica.
The authors took advantage of their previous studies on the identification of protein biomarkers connected to immune system activation, following a ten days intensive physical exercise, and extended their analysis to a larger period of time (28 weeks). They compared protein levels during normal and overstressing conditions with the purpose to collect a list of biomarkers useful to identify non-functional overreaching (NFOR) and overtraining syndrome (OTS) disturbances.
Authors performed an accurate analysis, detecting a large group of blood protein (31 up- and 35 down-regulated), mainly connected to the immune system and nutrition metabolic processes.
Results of their global proteomic approach were very impressive and of high quality.
On this basis, I consider the work very interesting and also significant in the field of studies of athletic performances. Moreover, the manuscript doesn’t need language revision.
For these reasons this could be considered a useful contribution in the context of proteomic research, and I believe that the above-mentioned manuscript is be suitable for publication on Proteomes.
Author Response
Thank you for taking the time and effort to review our paper. We appreciate your positive response.
Reviewer 3 Report
In this article, Nieman et al. go on the application of proteomics on the molecular characterization of dried blood spot (DBS) specimens, started with two their previous works. In the present paper, a similar workflow has been applied in biomarker discovering/confirmation of NFOR and OTS after a very severe training. From a proteomic point of view, this has meant a larger number of weekly samples of poor amount of plasma. Very sophisticated proteomic approach (DIA), instrumentation (Thermo Scientific Orbitrap Fusion Tribrid Mass Spectrometer) and software for data analysis (DIA-neural networks) have been employed. The results, even though largely confirmatory, are relevant in physical sciences.
However, since the target of the Proteomes journal is a specific community, in my opinion the article requires these minor improvements:
- over the entire paper, please delete the adjective "global" in front of "proteomics" (it is redundant), and, when appropriate, replace it with "plasma";
- In the Introduction (page 2 out 17): check if the citation of reference 5 (to the string paper) is correct;
- In the material section:
- a) in the "Proteomic Procedure" section, overall chromatographic elution condition/time has to be inserted;
- b) in the "data processing" section, describe the criterion by which "Retention times of the Proteomes library were adapted to instrument specific retention times" (MS2 spectra?);
- c) in the "statistical analysis" section: clarify if, and how many, "replicates" have been used in the LFQ analysis (are extracts from each DBS specimen been split in technical replicates? before or after trypsin digestion? Did you used as "replicates" the weekly collected samples encompassed into a selected period?) - A figure of the workflow should help readers in meaning this point;
- In table I and II the DIA identification scores are missed
Author Response
Over the entire paper, please delete the adjective "global" in front of "proteomics" (it is redundant), and, when appropriate, replace it with "plasma".
RESPONSE: We changed the verbage as you recommended throughout the paper.
In the Introduction (page 2 out 17): check if the citation of reference 5 (to the string paper) is correct.
RESPONSE: We fixed this reference. Thanks.
In the material section:
- a) in the "Proteomic Procedure" section, overall chromatographic elution condition/time has to be inserted;
“Flowrate was 300nl/min (buffer A, HPLC H2O, 0.1% formic acid; buffer B ACN, 0.1% formic acid; 60 min run method, 40 minute gradient: 0-3 min: 2% buffer B, 3-4 min: 2 to >5.5% buffer B, 4-9 min: 5.5 to >9.9% buffer B, 9-16 min: 9.9 to >15.1% buffer B, 16-17 min: 15.1 to >15.7% buffer B, 17-22 min: 15.7 to >19.5% buffer B, 22-23 min: 19.5 to >20.4% buffer B, 23-29 min: 20.4 to >25.1% buffer B, 29-30 min: 25.1 to >26% buffer B, 30-31 min: 26 to >26.8% buffer B, 31-32 min: 26.8 to >27.8% buffer B, 32-33 min: 27.8 to >28.5% buffer B, 33-34 min: 28.5 to >29.3% buffer B, 34-35 min: 29.3 to >30.4% buffer B, 35-36 min: 30.4 to >31.5% buffer B, 36-37 min: 31.5 to >32.9% buffer B, 37-38 min: 32.9 to >35% buffer B, 38-39 min: 35 to >37.5% buffer B, 39-40 min: 37.5 to >40% buffer B, 40-45 min: 95% buffer B, 45-60 min: 2% buffer B).”
RESPONSE: This is long and we opted for this description:
“Flowrate was 300nl/min (buffer A, HPLC H2O, 0.1% formic acid; buffer B ACN, 0.1% formic acid; 60 min run method, 40 minute gradient: 0-3 min: 2% buffer B, 3-40 min: nonlinear stepwise gradient from 2 to >40% buffer B, 40-45 min: 95% buffer B, 45-60 min: 2% buffer B)”
- b) in the "data processing" section, describe the criterion by which "Retention times of the Proteomes library were adapted to instrument specific retention times" (MS2 spectra?);
The retention time adjustment was performed using naturally occurring peptides in the iterative calibration approach. During the first pass, precursors are identified without RT adjustment using wide mass accuracy quadratic spline that approximate RT transformation. In successive steps both mass accuracy and spline are iteratively refined until convergence. The obtained RT transformation function is applied for the main processing part.
- c) in the "statistical analysis" section: clarify if, and how many, "replicates" have been used in the LFQ analysis (are extracts from each DBS specimen been split in technical replicates? before or after trypsin digestion? Did you used as "replicates" the weekly collected samples encompassed into a selected period?) - A figure of the workflow should help readers in meaning this point;
Every sample has been measured once so there were no technical replicates. To underline the quality of our measurements we monitored CV values of “pooled” samples (all samples mixed) before, after and during the measurements of the samples. We have added that sentence to the “Proteomics procedures” section.
“Pooled samples were run to monitor CV values and assess the quality of label-free quantitation.”
In table I and II the DIA identification scores are missed
For the analysis only peptides with an FDR of lower than 1% were taken into consideration. The peptide qvalue was set to be equal to the best precursor qvalue. Therefore, at this level any identification scores are not relevant.